# *MimicDreamer*: Aligning Human and Robot Demonstrations for Scalable VLA Training

## Abstract

Vision Language Action (VLA) models derive their generalization capability from diverse training data, yet collecting embodied robot interaction data remains prohibitively expensive. In contrast, human demonstration videos are far more scalable and cost-efficient to collect, and recent studies confirm their effectiveness in training VLA models. However, a significant domain gap persists between human videos and robot-executed videos, including unstable camera viewpoints, visual discrepancies between human hands and robotic arms, and differences in motion dynamics. To bridge this gap, we propose *MimicDreamer*, a framework that turns fast, low-cost human demonstrations into robot-usable supervision by jointly aligning vision, viewpoint, and actions to directly support policy training. For visual alignment, we propose H2R ALIGNER, a video diffusion model that generates high-fidelity robot demonstration videos by transferring motion from human manipulation footage. For viewpoint stabilization, EGOSTABILIZER is proposed, which canonicalizes egocentric videos via homography and inpaints occlusions and distortions caused by warping. For action alignment, we map human hand trajectories to the robot frame and apply a constrained inverse kinematics solver to produce feasible, low-jitter joint commands with accurate pose tracking. Empirically, VLA models trained purely on our synthesized human-to-robot videos achieve few-shot execution on real robots. Moreover, scaling training with human data significantly boosts performance compared to models trained solely on real robot data; our approach improves the average success rate by $14.7\%$ across six representative manipulation tasks.

## 1 Introduction

Vision Language Action (VLA) models (Black et al., 2024; 2025; X Square Robot Team, 2025; Cheang et al., 2025a; Bjorck et al., 2025) have shown strong generalization in robotic manipulation, but their progress is constrained by the cost and efficiency of data collection. Meanwhile, large-scale datasets (Khazatsky et al., 2025; Collaboration et al., 2025; AgiBot-World-Contributors et al., 2025) often rely on long teleoperation across heterogeneous hardware, which is time-consuming and limits task diversity. Unlike computer vision and natural language processing that can leverage Internet-scale corpora (Schuhmann et al., 2022; Dodge et al., 2021), robotics lacks cheap and abundant data sources. Human demonstrations (Bahl et al., 2022; Lepert et al., 2025; Grauman et al., 2022b) provide a more efficient and lower-cost path. Hand videos and action trajectories can be gathered quickly without continuous robot execution (Chao et al., 2021; Kwon et al., 2021), reducing hardware dependence and maintenance overhead. More importantly, human motion naturally encapsulates strategies and efficiencies observed in real operations, not brittle, preprogrammed paths, but adaptable procedures. Using human demonstrations as a primary data source, therefore, both reduces collection cost and supplies broadly applicable supervision for VLA training.

Existing mimic methods (Wang et al., 2023; Kareer et al., 2024; Xie et al., 2025; Yang et al., 2025a; Qiu et al., 2025) show that human demonstrations can effectively improve robot policy learning. Most of these methods incorporate human data as auxiliary signals or in limited pipelines, rather than turning them into fully robot-usable supervision for large-scale training. Human demonstrations cannot be used directly (Bahl et al., 2022; Kareer et al., 2024) because of domain and embodiment mismatches. We therefore convert human demonstrations into robot supervision and train VLA models end-to-end on the converted data. Direct transfer, however, still faces three gaps: view-

point, actions, and vision. (1) For the viewpoint, first-person human operation videos are typically captured by moving cameras with parallax and jitter, which complicates spatiotemporal alignment across sequences and tasks. (2) For actions, humans express intent through end-effector trajectories and dexterous manipulation, whereas robots operate in joint space under kinematic and dynamic constraints, making the semantics-to-control mapping often indirect and difficult to implement. (3) For vision, human hands and robot arms differ significantly in appearance, materials, and motion statistics, limiting the direct transfer of visual representations. Existing methods typically address only one of these issues (Kareer et al., 2024; Yang et al., 2025a), lacking a systematic approach that simultaneously tackles viewpoint stabilization, executable action mapping, and visual consistency.

Therefore, we propose *MimicDreamer*, a framework that turns fast, low-cost human demonstrations into robot-usable supervision by jointly aligning vision, viewpoint, and actions. To bridge the vision gap, we propose H2R ALIGNER, a video diffusion model that renders high-fidelity robot-arm videos by transferring motion from human manipulation footage while respecting arm geometry and camera priors (Yang et al., 2025c). Quantitative and qualitative results show realistic arm appearance and contact geometry consistent with the source task. For viewpoint stabilization, EGOSTABILIZER canonicalizes egocentric frames via homography-based warping to a task-level reference view (estimated by averaging per-category rotations) and inpaints distortions or occlusions introduced by warping (Zhou et al., 2023). Experiment results confirm reduced ego-motion drift and improved cross-sequence comparability. To align the action space, we encode intention as relative end-effector pose increments in the shared frame and execute it via a constrained inverse kinematics (IK) solver with distributional normalization and temporal smoothness, yielding feasible, low-jitter joint trajectories. Visualized rollouts exhibit accurate end-effector tracking while respecting joint and velocity limits.

In experiments, training the VLA model (Black et al., 2024) solely on *MimicDreamer*-synthesized human to robot videos enables few-shot execution on real robots. Across six representative manipulation tasks, increasing the scale of human demonstration data yields consistent gains, improving an average success rate by 14.7% over a baseline trained only on real robot data. The primary contributions of this work are as follows:

1. We propose *MimicDreamer*, a unified human–robot egocentric demonstrations transferring framework that simultaneously reduces the human-to-robot discrepancy along vision, viewpoint, and action dimensions and enables scalable VLA training from low-cost human demonstrations.

2. For vision, we introduce H2R ALIGNER based on video diffusion and geometric camera priors to synthesize high-fidelity robot arm videos. For viewpoint, we introduce EGOSTABILIZER, which canonicalizes frames to a task reference view by homography and repairs warping occlusions. For actions, we map human hand trajectories to the robot frame and apply constrained IK to produce feasible, low-jitter joint commands with accurate pose tracking.

3. The VLA policy trained on synthesized robot demonstrations achieves few-shot execution on real robots, and across six manipulation tasks, we realize scalable VLA training, improving an average success rate over the robot data baseline by 14.7%, demonstrating both stronger generalization and higher sample efficiency.

## 2 RELATED WORK

### 2.1 VISION LANGUAGE ACTION MODELS

Recent Vision Language Models (VLM) (Li et al., 2023; Bai et al., 2025; Group, 2025) have made rapid progress in grounding and instruction following, providing strong semantic priors for downstream control (Huang et al., 2024; Kuo et al., 2023). Building on these foundations, Vision Language Action (VLA) models (Sapkota et al., 2025) aim to couple internet-scale vision-language semantics with control policies, mapping observations and natural-language instructions directly to executable actions in embodied settings. Early pioneering works demonstrated this potential; for instance, SayCan (Ahn et al., 2022) combined a Large Language Model (LLM) for high-level reasoning with learned affordance functions to ground feasible skills, while PaLM-E (Driess et al., 2023) injected continuous sensory tokens into an LLM, and RT-2 (Brohan et al., 2023) showed that web-scale vision-language pretraining can transfer semantic knowledge into action policies. A

prominent trend is the adoption of dual-system or hierarchical frameworks that separate high-level planning from low-level execution. This approach is exemplified by models like Galaxea's G0 (Jiang et al., 2025), GR00T N1 (NVIDIA et al., 2025), and $\pi_{0.5}$ (Cheang et al., 2025b), which use a VLM as a deliberative planner to interpret scenes and decompose tasks into sub-goals. In contrast, other works focus on creating more tightly integrated, end-to-end models. WALL-OSS (Zhai et al., 2025), for instance, directly confronts the modality and training objective gaps between VLM and robotics by introducing a tightly-coupled mixture of experts architecture and a unified cross-level chain of thought framework that seamlessly unifies reasoning, planning, and action synthesis.

To achieve open-world generalization, recent works augment robot-specific datasets by co-training on heterogeneous data sources. Models like $\pi_{0.5}$ (Intelligence et al., 2025) have demonstrated the benefits of a mixed training recipe including web data, cross-embodiment trajectories, and verbal instructions. This concept is further structured by GR-3 (Cheang et al., 2025b) and GR00T N1 (NVIDIA et al., 2025), which utilize a "data pyramid" of web, synthetic, and real-robot data. Despite these advances, the scarcity of robot data remains a primary bottleneck. *MimicDreamer* addresses this by leveraging abundant egocentric videos to enhance policy learning.

## 2.2 LEARNING FROM EGOCENTRIC VIDEOS

Egocentric videos have emerged as a scalable supervision source for robotic arms, offering a cost-effective alternative to extensive robot teleoperation (Nair et al., 2022; Bahl et al., 2023; Wang et al., 2023; Kareer et al., 2024; Yang et al., 2025b). Early works in this area leveraged large-scale human video datasets primarily for perception-centric pre-training. For instance, R3M (Nair et al., 2022) pretrains a frozen visual encoder on Ego4D (Grauman et al., 2022a) using time-contrastive and video-language objectives, which improves the data efficiency of downstream policy learning. Similarly, VRB (Bahl et al., 2023) learns to extract visual affordances, identifying how to interact, from human videos on the Internet to guide various control and reinforcement learning paradigms.

Building upon these perceptual priors, subsequent research has focused on more direct imitation from human behaviour, translating first-person demonstrations into robot-executable policies. MimicPlay (Wang et al., 2023) adopts a hierarchical strategy, learning a high-level latent plan from unstructured "human play" data to guide a low-level visuomotor controller. In contrast, EgoMimic (Kareer et al., 2024) proposes a unified framework that co-trains a single policy on both egocentric human videos and robot data. Further advancing the direct use of human data, EgoVLA (Yang et al., 2025b) pre-trains a VLA model exclusively on human videos to predict human wrist and hand actions; these actions are then mapped to the robot's control space via inverse kinematics and retargeting, followed by a final fine-tuning stage on robot data to refine the policy.

While these methods address individual aspects of the human-robot gap, they do not offer a holistic solution. Our work, *MimicDreamer*, introduces a framework that systematically aligns human and robot data across three critical dimensions simultaneously: vision, viewpoint, and actions. By effectively turning human videos into robot-usable supervision, our framework not only enables few-shot adaptation but also demonstrates that performance scales consistently as more human data is added, significantly boosting success rates over baselines trained only on robot data.

## 3 METHOD

As shown in Figure 1, we propose *MimicDreamer*, a low-cost pipeline that turns egocentric human demonstrations into robot-usable supervision through viewpoint canonicalization, human-to-robot visual alignment, and action alignment. Given egocentric videos, EGOSTABILIZER applies warp perspective and background inpainting to produce stable egocentric videos. In parallel, 3D hand trajectories are mapped to the robot frame and converted into feasible, low-jitter robot actions via a constrained IK solver. Then the robot actions, together with the robot URDF, drive the manipulator motion in the simulation engine, and a calibrated virtual camera with preset intrinsics and extrinsics renders egocentric simulation robot videos, which we use as robot-view priors. H2R ALIGNER consumes the stable egocentric and rendered simulation robot videos to synthesize paired robot-view manipulation videos. We then train a VLA policy on aligned synthesized robot videos and IK-derived actions, using a few real robot data for grounding, thereby enabling robot-policy learning directly from egocentric human demonstrations.

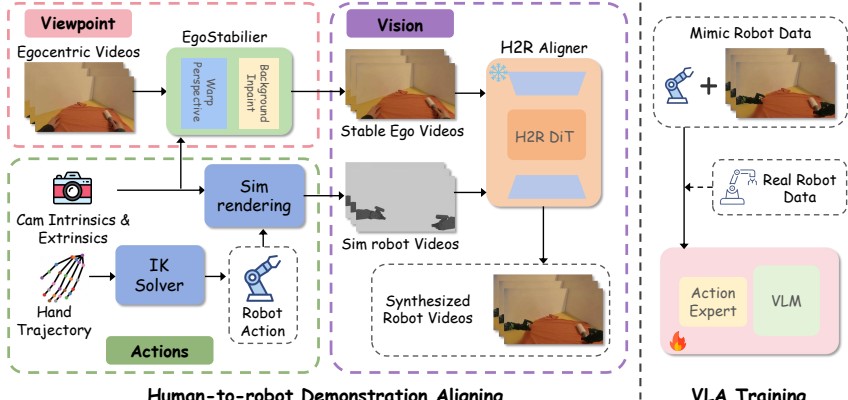

Figure 1: Overview of *MimicDreamer*. Viewpoint branch (top left): egocentric videos are stabilized by EGOSTABILIZER (warp perspective + background inpainting) to produce stable egocentric videos. Camera intrinsics/extrinsics and the robot URDF drive sim rendering to generate additional stable ego views. Action branch (bottom left): 3D hand trajectories are converted to robot actions with IK solver. Visual alignment (right): H2R ALIGNER learns to bridge the human-to-robot visual gap using stable egocentric videos and simulation robot videos. The resulting synthesized robot videos and robot actions are used for VLA training.

## 3.1 VIEWPOINT STABILIZATION

Egocentric videos often contain nonstationary camera motion such as head micro-shake, rapid swings, and scale changes. An unstable background enlarges the appearance gap between robot-view priors and human videos, which weakens the effectiveness of using rendered priors to guide the synthesis of robot-view videos. We therefore propose EGOSTABILIZER. By stabilizing and canonicalizing the viewpoint, it reduces inter-frame angular variation and high-frequency jitter, improves registration robustness and alignment quality, increases data efficiency, and provides cleaner supervision for subsequent H2R visual alignment and VLA training.

**Warp Perspective** We match features between adjacent frames or against a reference frame and estimate a homography $H_t$ with RANSAC (Fischler & Bolles, 1981; Hartley & Zisserman, 2004). The camera path is temporally smoothed (Liu et al., 2011) to obtain $\tilde{H}_t$, and we form a compensation transform $W_t = \tilde{H}_t H_t^{-1}$. Applying this compensation to each frame removes high-frequency jitter and aligns frames to a canonical camera path:

$$\tilde{I}_t(\mathbf{x}) = I_t\big((W_t)^{-1}\mathbf{x}\big), \tag{1}$$

where $\mathbf{x}$ denotes a pixel in homogeneous coordinates, $I_t$ is the original frame, and $\tilde{I}_t$ is the stabilized but potentially holey frame. We then compute the maximal common visible region over all $\{\tilde{I}_t\}$ and apply uniform scaling and light cropping to avoid black borders and field-of-view jitter.

**Video Inpainting** From out-of-bounds and interpolation-missing regions after remapping, we derive a binary mask $M_t$. The stabilized frames $\{\tilde{I}_t\}$ together with $\{M_t\}$ are fed into video inpainter model (Zhou et al., 2023), which uses spatiotemporal feature propagation and cross-frame consistency to aggregate reliable observations from neighboring frames and to fill holes and disocclusions, producing $\hat{I}_t$ with coherent backgrounds and smooth boundaries. This step alleviates artifacts due to geometric compensation, reduces temporal flicker, and yields a stabilized sequence that is better suited for H2R visual alignment and the synthesis of robot-view videos.

## 3.2 ACTIONS ALIGNMENT

We construct a unified H2R action space that deterministically maps human wrist poses to robot joint commands while respecting kinematics and smoothness. For a bimanual robot, the action is

$$\mathbf{q}_t = \begin{bmatrix} \mathbf{q}_t^L \\ \mathbf{q}_t^R \end{bmatrix}, \quad \mathbf{q}_t^a \in \mathbb{R}^7 \ (a \in \{L, R\}), \quad \mathbf{q}_t^a = \big[q_{t,1}^a, \ldots, q_{t,6}^a, \ g_t^a\big]^\top, \tag{2}$$

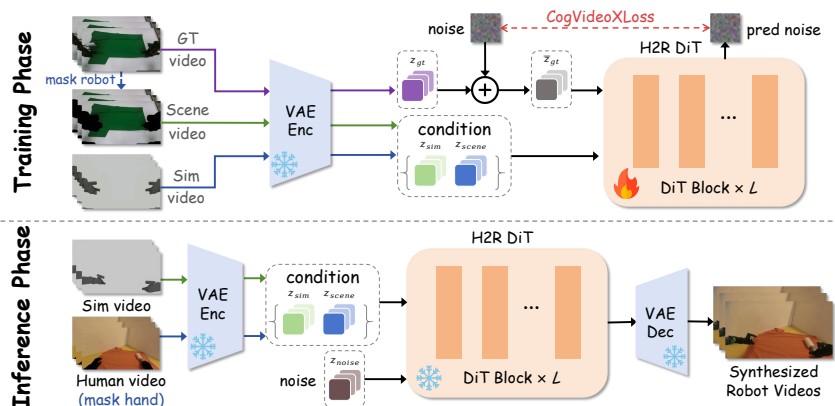

Figure 2: H2R ALIGNER. During training, the real robot video $V_{\text{gt}}$, background $V_{\text{scene}}$, and simulated foreground $V_{\text{sim}}$ are encoded by a frozen VAE and channel-concatenated as $[\tilde{z}_{\text{tar}}, z_{\text{scene}}, z_{\text{sim}}]$ before entering the trainable H2R DiT, optimized with CogVideoXLoss loss. During inference, a hand-masked human background and IK-replayed simulation serve as conditions; the target starts from noise, is denoised by H2R DiT, and decoded by the frozen VAE into synthesized robot videos.

where $t$ is time, the first six entries control the End-Effector (EE) pose, and $g_t^a$ is the gripper DoF.

**Human-side Normalization**  We express human 3D keypoints in a body-centric frame $\mathcal{F}_B$: $\mathbf{p}^B = \mathbf{R}_B^\top(\mathbf{p} - \mathbf{o}_B)$, and estimate a continuous wrist pose $(\mathbf{p}_t^{H,B}, \mathbf{R}_t^{H,B})$ from the hand skeleton. We then register to the robot base $\mathcal{F}_R$ via a rigid transform $(\mathbf{R}_{HR}, \mathbf{t}_{HR})$:

$$\mathbf{p}_t^* = \mathbf{R}_{HR}\mathbf{p}_t^{H,B} + \mathbf{t}_{HR}, \qquad \mathbf{R}_t^* = \mathbf{R}_{HR}\mathbf{R}_t^{H,B}. \tag{3}$$

**Orientation Treatment**  Because the human wrist behaves like a near-spherical joint while many EEs largely roll around the tool axis, we align only the tilt (pitch/yaw). The process of softly masking roll can be represented as:

$$\phi(\mathbf{q}) = \text{Log}\big(\mathbf{R}_t^* \, \mathbf{R}_{\text{EE}}(\mathbf{q})^\top\big)^\vee \in \mathbb{R}^3, \quad \mathbf{W}_R = \text{diag}(w_x, w_y, w_z), \; w_z \ll w_x, w_y. \tag{4}$$

**IK Resolver**  For each arm $a \in \{L, R\}$, we recover a feasible joint configuration by solving

$$\min_{\mathbf{q}^a} \; \big\|\mathbf{p}_{\text{EE}}(\mathbf{q}^a) - \mathbf{p}_t^{*a}\big\|_2^2 + \phi(\mathbf{q}^a)^\top \mathbf{W}_R \phi(\mathbf{q}^a) + \lambda\big\|\mathbf{q}^a - \mathbf{q}_{t-1}^a\big\|_2^2 \quad \text{s.t. } \mathbf{q}_{\min} \le \mathbf{q}^a \le \mathbf{q}_{\max}. \tag{5}$$

We warm-start from $\mathbf{q}_{t-1}^a$ and use Damped Least Squares (DLS) (Buss & Kim, 2005) steps for fast, smooth trajectories. The DLS update, Jacobian forms, stopping criteria, and ablations are provided in the Appendix A.

**Gripper**  The binary gripper command $g_t^a \in [0, 1]$ is inferred from hand openness via a lightweight VGG-based (Simonyan & Zisserman, 2015) classifier and a short median filter reduces flicker.

### 3.3 VISUAL ALIGNMENT

During experiments, we found that, due to the large visual discrepancy between the PiPER manipulator and human hands, training a VLA policy with first-person human demonstration videos plus aligned actions struggled to accomplish the corresponding tasks. To remove this human–robot visual gap, we design H2R ALIGNER as shown in Figure 2, a unified visual aligner from human to robot that converts inexpensive but "not directly usable" human clips into robot training samples that are executable, evaluable, and semantically consistent. Built on CogVideoX-5b-I2V (Hong et al., 2022; Yang et al., 2025c), H2R ALIGNER conditions on instruction embeddings, the real video stream, and the simulation rendering stream, trains a multi-conditional video diffusion generator, and uses the generated clips to construct the mimic robot dataset for subsequent VLA post-training.

During the training phase, we organize batches at the episode level with length $f$. Each episode $e$ contains a joint-position sequence and action labels $q, a \in \mathbb{R}^{f \times 14}$, as well as a head-camera video

of the real manipulator $\mathbf{V}_{\mathrm{gt}} \in \mathbb{R}^{f \times H \times W \times c}$. We decompose the conditioning inputs into two parts: a robotic foreground stream and a background scene stream. The foreground stream is obtained by a simulation replay of the real joint trajectory. Given the URDF $u_r$ that matches the real platform and a virtually calibrated camera (intrinsics and extrinsics aligned to real setup), the simulator renders:

$$\mathbf{V}_{\mathrm{sim}} = \mathrm{Sim}(q, u_r). \tag{6}$$

The background stream provides environmental observations without the manipulator. To this end, we project the simulator's manipulator silhouette into the real video to obtain a mask, apply slight dilation to mitigate boundary pixels, and remove the masked region from $\mathbf{V}_{\mathrm{gt}}$ to obtain a clean background sequence $\mathbf{V}_{\mathrm{scene}} \in \mathbb{R}^{f \times H \times W \times c}$. We use three videos to train H2R ALIGNER $\{\mathbf{V}_{\mathrm{gt}}, \mathbf{V}_{\mathrm{scene}}, \mathbf{V}_{\mathrm{sim}}\}$. Here $\mathbf{V}_{\mathrm{gt}}$ is used only as the target path for noising/denoising during training, while $\mathbf{V}_{\mathrm{scene}}, \mathbf{V}_{\mathrm{sim}}$ serve as conditional paths. They are encoded by a shared, frozen video VAE into latent sequences $\{z_{\mathrm{gt}}, z_{\mathrm{scene}}, z_{\mathrm{sim}}\}$. The target latent $z_{\mathrm{tar}}$ is perturbed at a randomly sampled diffusion timestep to produce the denoising target $\tilde{z}_{\mathrm{tar},t}$; the scene and simulated-foreground latents remain clean as conditions. We then concatenate the three along the channel dimension and, together with 3D spatiotemporal positional encodings, feed them into H2R DiT to perform latent-space denoising and conditional fusion:

$$\tilde{z}_{\mathrm{tar},t} = \sqrt{\bar{\alpha}_t}\, z_{\mathrm{tar}} + \sqrt{1 - \bar{\alpha}_t}\, \epsilon, \qquad \epsilon \sim \mathcal{N}(0, \mathbf{I}), \tag{7}$$

$$z_t = \mathrm{concat}_{\mathrm{channels}}\big[\tilde{z}_{\mathrm{tar},t},\, z_{\mathrm{scene}},\, z_{\mathrm{sim}}\big], \tag{8}$$

where $\bar{\alpha}_t$ is the cumulative product of the noise-schedule coefficients up to timestep $t$.

Next, the H2R DiT denoises $z_t$ in latent space under 3D spatio-temporal positional encodings, outputs the residual prediction in timestep $t$, and updates the H2R DiT backbone during training. Let $\theta$ denote the trained H2R DiT parameters, the final optimized latent is:

$$z_{\mathrm{tar},0} = T_\theta\big(z_{\mathrm{scene}}, z_{\mathrm{robot}}; \xi, \tau\big), \tag{9}$$

During the inference phase, the foreground stream replays the IK-derived joint sequence $q^{\mathrm{ik}}$ in simulation to produce $\mathbf{V}_{\mathrm{sim}}^{\mathrm{ik}}$. The background stream uses the real hand video segmented by Grounded-SAM2 (Ravi et al., 2024; Ren et al., 2024), with slight dilation to obtain the hand mask, yielding $\mathbf{V}_{\mathrm{scene}}^{\mathrm{ik}}$; the human video is first stabilized by the viewpoint procedure in Sec. 3.1 before entering this module. The target latent is initialized from noise $\xi$, and $\mathbf{V}_{\mathrm{gt}}$ is not used at inference.

Finally, we create the mimic robot dataset by time-aligning the synthesized robot videos ($V_{\mathrm{rob}}$) with their corresponding actions ($a^{\mathrm{ik}}$). This dataset translates human strategies into the robot's visual domain and can be used independently for policy training or combined with real robot data to improve robustness. By preserving human strategy while constraining visual appearance to the robot's domain, H2R ALIGNER transforms inexpensive human videos into executable and semantically aligned training samples. This provides a stable data foundation for learning instruction-to-control mappings.

## 3.4 VLA TRAINING

We use the mimic robot data synthesized by H2R ALIGNER and IK solver as the primary training source and then mix in a small amount of real demonstrations for post-training, so the policy attains both broad semantic alignment and real-world executability. We initialize the policy from the $\pi_0$ pretrained model (Black et al., 2024), reusing its VLM backbones and action tokenization, and perform post-training on our data. During training, the instruction is encoded by a text encoder to obtain an instruction embedding, and a short-window video encoder processes the video. The policy head outputs intention-level controls, which are projected to joint commands, ensuring feasible, low-jitter trajectories. We supervise the action tokens with a conditional flow matching objective (Lipman et al., 2022; Tong et al., 2024):

$$\mathcal{L}_{\mathrm{CFM}}(\theta) = \mathbb{E}_{c,\mathbf{a},t,\epsilon}\left[\|\mathbf{u}_\theta(\mathbf{y}_t, c, t) - \mathbf{u}^\star(\mathbf{y}_t \mid \mathbf{a}, \epsilon, t)\|_2^2\right], \tag{10}$$

where $\theta$ are the model parameters, $c$ is the fused context from the video and instruction encoders, $\mathbf{a} \in \mathbb{R}^d$ is the ground-truth action token, $t \sim \mathcal{U}(0,1)$ and $\epsilon \sim \mathcal{N}(\mathbf{0}, \mathbf{I})$, the noisy interpolant is $\mathbf{y}_t = \alpha(t)\mathbf{a} + \sigma(t)\epsilon$ with schedules $\alpha(0) = 0$, $\alpha(1) = 1$, $\sigma(1) = 0$, the target velocity is $\mathbf{u}^\star(\mathbf{y}_t \mid \mathbf{a}, \epsilon, t) = \dot{\alpha}(t)\mathbf{a} + \dot{\sigma}(t)\epsilon$, and $\mathbf{u}_\theta(\cdot)$ is the learned velocity predictor. We optimize $\theta$ with AdamW (Loshchilov & Hutter, 2019) and select the final checkpoint by validation of CFM loss $\mathcal{L}_{\mathrm{CFM}}$.

Table 1: Quantitative Results Across Three Training Setups. SR and PSR for a Robot Only baseline (20 robot data), w. Minimal Robot trained primarily on synthesized data (20 human-to-robot data + 3 robot data), and w. Equal Data using a balanced mix (20 human-to-robot data + 20 robot data).

| Method | Pick Bag | | Clean Surface | | Stack Bowls | | Dry Hands | | Insert Tennis | | Stack Cups | |
|---|---|---|---|---|---|---|---|---|---|---|---|---|
| | SR↑ | PSR↑ | SR↑ | PSR↑ | SR↑ | PSR↑ | SR↑ | PSR↑ | SR↑ | PSR↑ | SR↑ | PSR↑ |
| Robot Only | 70% | 82% | 90% | 90% | 65% | 80% | 80% | 88% | 25% | 38% | 65% | 80% |
| w. Minimal Robot | 75% | 85% | 95% | 95% | 70% | 85% | 85% | 93% | 25% | 43% | 65% | 85% |
| w. Equal Data | **90%** | **93%** | **100%** | **100%** | **90%** | **93%** | **100%** | **100%** | **45%** | **70%** | **90%** | **90%** |

## 4 EXPERIMENTS

### 4.1 RESULTS OF VLA POLICY ON MIMIC ROBOT DATA

**Experiment Setup** In this study, we employ the EgoDex dataset (Hoque et al., 2025) for our experiments, which provides a large-scale collection of egocentric videos. The EgoDex dataset is essential for training models to learn dexterous manipulation, offering 829 hours of high-quality, 1080p egocentric videos paired with 3D upper-body poses for 194 tasks.

**Evaluation Tasks** To evaluate our framework's ability to generalize from human demonstrations to robotic actions, we constructed six scenarios that resemble those in the EgoDex dataset, e.g., Pick Bag, Clean Surface, Stack Bowls, Dry Hands, Insert Tennis, and Stack Cups. The specific task and subtask setup are detailed in the Appendix B.3.

**Evaluation Metrics** Following (Yang et al., 2025b), the evaluation is conducted using Success Rate (SR), which quantifies overall task success, and Progress Success Rate (PSR), which measures the average number of completed subtasks relative to the total subtasks in each task.

### 4.1.1 FEW-SHOT EXPERIMENTAL RESULTS

We conducted experiments with three distinct data configurations to evaluate the effectiveness of the *MimicDreamer* framework in improving robotic task execution. As shown in Table 1, we present the performance of each experimental setup across six manipulation tasks. Averaged over all tasks, the Robot Only setup attains 65.8% SR/76.3% PSR, whereas the w. Minimal Robot setup already lifts performance to 70.0%/81.0%. The strongest results come from the Equal Data setup: 85.0%/91.0%. Per-task, the Equal Data method improves SR on every task ($+10 \sim 25\%$) and PSR on every task ($+10 \sim 32\%$), achieving 100% SR/PSR on Clean Surface and Dry Hands. The largest relative gains appear on the hardest setting, the performance on Insert Tennis Task grows from 25%/38% to 45%/70% and on long-horizon stacking tasks (Stack Bowls: +20%SR/+13%PSR; Stack Cups: +25%SR/+10%PSR).

Even with minimal robot data, *MimicDreamer* surpasses Robot Only setup on both metrics and most tasks, indicating that human demonstrations provide strong priors that transfer to robot control. The average gap between SR and PSR shrinks from 10.5% (Robot Only) to 6.0% (w. Equal Data), and PSR variability across tasks drops. Together, these trends suggest that *MimicDreamer* converts more partial attempts into full successes and behaves more consistently across diverse tasks.

### 4.1.2 SCALING EXPERIMENT RESULTS

To assess the scalability of the Mimic Robot Data, we start from a baseline VLA trained with 20 real-robot trajectories and then progressively add human-to-robot data from 5 to 30. As the number of human demonstrations increases, both SR and PSR rise monotonically across all six tasks, showing that robot demonstrations synthesized from human demonstrations by *MimicDreamer* exhibit clear scalability in VLA training. As shown in Figure 3, the largest gains occur between 5 and 20 human data, after which improvements exhibit diminishing returns due to ceiling effects as success rates approach 100%, indicating a fast-then-steady scaling trend. At a 50–50 mix percentage of human-to-robot and robot data (20 human + 20 robot), the success rate improves over the baseline by 11.0%, 10.0%, 13.0%, 12.0%, 32.0%, and 10.0% across the six tasks. Overall, viewpoint canonicalization and visual alignment first deliver stable partial success gains, while constrained IK with

temporal smoothing converts partial success into complete task success; once visual and viewpoint factors saturate, remaining headroom is dominated by dexterous skills such as precise grasping, which benefit more from additional human demonstrations. More quantitative results are shown in Appendix B.6.

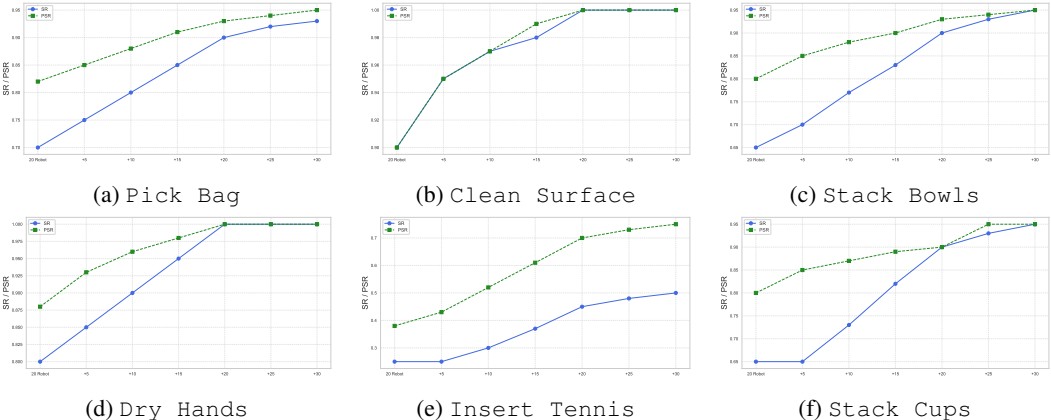

(a) `Pick Bag`   (b) `Clean Surface`   (c) `Stack Bowls`

(d) `Dry Hands`   (e) `Insert Tennis`   (f) `Stack Cups`

Figure 3: Scaling Experiment Results. As more human-to-robot data is added, the *MimicDreamer*'s success rate monotonically increases across all six tasks.

## 4.2 RESULTS OF H2R ALIGNER

**Experiment Setup**   We train H2R-Aligner on 24 manipulation categories. Raw clips are randomly cropped to $640 \times 360$ and resized to $672 \times 384$; this yields $3,735$ samples, each 64 frames at 30 fps, split $9:1$ into train and val set.

**Visual Results**   We present several visual results for H2R-ALIGNER on cloth manipulation. As shown in Figure 4, the top row is the original human demonstration, the middle row is the simulated replay with the same trajectories, and the bottom row is the synthesized robot-domain video. The results show that H2R-ALIGNER generates realistic robot-arm sequences aligned with both task semantics and background context. Additional examples are provided in Appendix B.5.

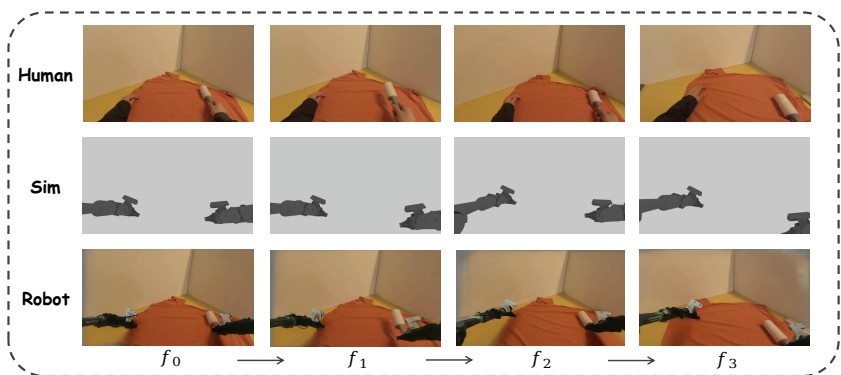

Figure 4: Visual Results of H2R ALIGNER. Top: original human demonstration video. Middle: replayed robot simulation from the same action trajectories. Bottom: synthesized robot-domain video generated by H2R ALIGNER. The generated sequences transfer human motions into robot-arm appearances while preserving background context and manipulation semantics.

## 4.3 RESULTS OF EGOSTABILIZER

**Quantitative Results**   To contextualize the following results, we adopt a unified evaluation protocol across six data categories, using the original videos as the reference. We report three stability

Table 2: Per-category, frame-weighted means. "↓" lower is better; "↑" higher is better. Cells show before → **after** (relative Δ%).

| Category | Videos | Stability ↓ | Jitter RMS ↓ | H-RMSE ↓ |
|---|---|---|---|---|
| Pick Bag | 332 | 0.4086 → **0.3752**(−8.2%) | 0.9757 → **0.8566**(−12.2%) | 0.00233 → **0.00166**(−28.9%) |
| Clean Surface | 1941 | 0.1144 → **0.0939**(−17.9%) | 0.1538 → **0.1421**(−7.6%) | 0.000568 → **0.000560**(−1.5%) |
| Stack Bowls | 2731 | 0.1156 → **0.0949**(−17.9%) | 0.1404 → **0.1321**(−5.9%) | 1.2245 → **1.2066**(−1.5%) |
| Dry Hands | 2681 | 0.4347 → **0.2952**(−32.1%) | 0.5777 → **0.4462**(−22.8%) | 1.0319 → **1.0040**(−2.7%) |
| Insert Tennis | 279 | 0.1065 → **0.0941**(−11.6%) | 0.2130 → **0.2030**(−4.7%) | 4.9562 → **4.8813**(−1.5%) |
| Stack Cups | 976 | 0.4369 → **0.3483**(−20.3%) | 1.3137 → **1.0448**(−20.5%) | 11.3364 → **10.6954**(−5.7%) |
| All | 8940 | -21.9% | -13.1% | -3.3% |

metrics, Stability (Grundmann et al., 2011), Jitter RMS (Liu et al., 2013), and Homography RMSE (H-RMSE) (Wang et al., 2019; Balntas et al., 2017) , to jointly assess viewpoint steadiness and geometric alignment. These complementary metrics quantify the impact of EGOSTABILIZER on stability and geometric consistency. Formal definitions and equations are provided in Appendix B.4.

As shown in Table 2, EGOSTABILIZER substantially enhances viewpoint consistency while preserving geometric fidelity. On average across all categories, our method reduces the Stability mean by 21.9% and the Jitter RMS by 13.1%, indicating a significant reduction in camera shake. This stabilization is achieved at a low geometric cost, evidenced by a modest 3.3% decrease in H-RMSE. A per-category analysis reveals that the stabilization gains are positively correlated with the initial instability of the sequence. For example, Dry Hands benefits the most (32.1% Stability reduction), whereas already-stable sequences such as Stack Bowls show more moderate improvements.

**Quality Result**   To isolate and evaluate viewpoint stabilization, we first remove dynamic objects by segmenting human hands and inpainting the background. This process yields background-only sequences where inter-frame changes are dominated by camera motion. As shown in Figure 5, we compare keyframes from a 300-frame sequence by tracking the displacement of static background features. The original video exhibits pronounced jitter, whereas keypoints in the EGOSTABILIZER output show negligible displacement, demonstrating robust viewpoint stabilization. Additional examples are provided in the supplementary materials.

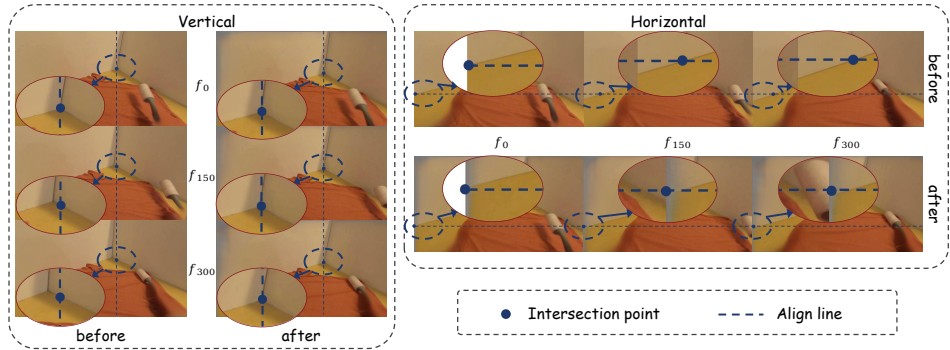

Figure 5: Qualitative evaluation of EGOSTABILIZER. On a 300-frame Clean Surface video, frames at indices 0, 150, and 300 are shown beforeand after stabilization. Keypoints such as wall corners and table–image intersections exhibit large jitter in the original video, whereas the stabilized outputs show negligible displacement, confirming effective viewpoint stabilization.

## 5 CONCLUSION

*MimicDreamer* converts low-cost human demonstrations into effective robot supervision by aligning visual content, viewpoint, and actions. Training VLA models on the transferred dataset enables few-shot execution on real robots and scales with additional human data. The approach lowers data collection cost while preserving generalization. Future work will target richer, dexterous, and deformable object manipulation, integrate force and contact cues, improve long-horizon temporal coherence, and expand cross-robot and cross-scene generalization. We also plan cost-aware scheduling of human and robot data and larger-scale synthesis to raise the ceiling of scalable VLA training.

## REPRODUCIBILITY STATEMENT

We have made every effort to ensure the reproducibility of our work. Figure 1 and Figure 2 clearly describe the proposed *MimicDreamer* framework and its three modules (EGOSTABILIZER, H2R-ALIGNER, and action alignment). The training details for both H2R-ALIGNER and the VLA policy are provided in the Appendix B.1, including model architectures, hyperparameters, and dataset preprocessing steps. Complete mathematical formulations of the objectives and metrics are also included in Appendix B.4. For experimental reproducibility, we describe data splits, training settings, and evaluation protocols in Section 4, and further report metric definitions in the supplementary materials. We plan to release our codebase, training scripts, and dataset processing pipeline to the community in the near future to further facilitate verification and extension of our work.

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

APPENDIX

## A   DETAILS FOR UNIFIED HUMAN-TO-ROBOT ACTION SPACE

**Human-side Coordinate Normalization**    All human 3D keypoints are expressed in a body-centric frame $\mathcal{F}_B$ whose origin is the spine base:

$$\mathbf{p}^B = \mathbf{R}_B^\top(\mathbf{p} - \mathbf{o}_B). \tag{11}$$

From these keypoints, we estimate a continuous wrist pose $(\mathbf{p}_t^{H,B}, \mathbf{R}_t^{H,B})$; $\mathbf{R}_t^{H,B}$ is constructed from stable anatomical axes (optionally using the mean of several metacarpophalangeal joints to reduce jitter).

**Human-to-robot Rigid Alignment**    Given the robot base frame $\mathcal{F}_R$, we use a fixed rigid transform $(\mathbf{R}_{HR}, \mathbf{t}_{HR})$ to place human motion in the robot workspace:

$$\mathbf{p}_t^* = \mathbf{R}_{HR}\mathbf{p}_t^{H,B} + \mathbf{t}_{HR}, \qquad \mathbf{R}_t^* = \mathbf{R}_{HR}\mathbf{R}_t^{H,B}. \tag{12}$$

**Tilt-only Orientation Treatment**    Instead of enforcing full SO(3) alignment, we emphasize palm *tilt* (pitch/yaw) and de-emphasize tool-axis roll. Let

$$\mathbf{R}_{\mathrm{err}}(\mathbf{q}) = \mathbf{R}_t^* \mathbf{R}_{\mathrm{EE}}(\mathbf{q})^\top, \qquad \boldsymbol{\phi}(\mathbf{q}) = \mathrm{Log}\big(\mathbf{R}_{\mathrm{err}}(\mathbf{q})\big)^\vee \in \mathbb{R}^3, \tag{13}$$

and apply a diagonal weight $\mathbf{W}_R = \mathrm{diag}(w_x, w_y, w_z)$ with $w_z \ll w_x, w_y$ to softly mask the roll channel.

**Per-arm IK Objective With Smoothness and Limits**    For each arm $a \in \{L, R\}$ we recover a feasible joint configuration by solving

$$\min_{\mathbf{q}^a} \; \big\|\mathbf{p}_{\mathrm{EE}}(\mathbf{q}^a) - \mathbf{p}_t^{*a}\big\|_2^2 + \boldsymbol{\phi}(\mathbf{q}^a)^\top \mathbf{W}_R \, \boldsymbol{\phi}(\mathbf{q}^a) + \lambda\big\|\mathbf{q}^a - \mathbf{q}_{t-1}^a\big\|_2^2 \tag{14}$$

$$\text{s.t. } \mathbf{q}_{\min} \leq \mathbf{q}^a \leq \mathbf{q}_{\max}. \tag{15}$$

We warm-start with $\mathbf{q}_{t-1}^a$ to encourage temporal smoothness and faster convergence. We implement the above with DLS steps on the stacked task error

$$\mathbf{e}(\mathbf{q}) = \begin{bmatrix} \mathbf{p}_{\mathrm{EE}}(\mathbf{q}) - \mathbf{p}_t^* \\ \mathbf{W}_R^{1/2}\, \boldsymbol{\phi}(\mathbf{q}) \end{bmatrix}, \tag{16}$$

$$\Delta\mathbf{q} = \mathbf{J}^\top\big(\mathbf{J}\mathbf{J}^\top + \mu^2\mathbf{I}\big)^{-1}\mathbf{e}(\mathbf{q}) \; - \; \lambda\big(\mathbf{q} - \mathbf{q}_{t-1}\big), \tag{17}$$

where $\mathbf{J}$ is the geometric Jacobian at $\mathbf{q}$ and $\mu$ is the damping coefficient. We iterate the update and enforce box constraints at each step:

$$\mathbf{q} \; \leftarrow \; \mathrm{clip}\big(\mathbf{q} + \Delta\mathbf{q}, \; \mathbf{q}_{\min}, \; \mathbf{q}_{\max}\big), \tag{18}$$

until the solution converges or a fixed, small number of steps is reached.

**Gripper**    For the seventh DoF, we infer a binary open/close command $g_t^a \in [0, 1]$ from the human hand state. A lightweight VGG-based classifier predicts the state from hand images; after manual spot-check correction on a small subset, we threshold to obtain $g_t^a$ and optionally apply a short median filter to reduce flicker.

# B    EXPERIMENT DETAILS

## B.1    HYPERPARAMETER SETTINGS

**H2R ALIGNER**    We train H2R ALIGNER on 24 manipulation categories. Raw clips (approximately $640 \times 460$) are randomly cropped to $640 \times 360$ and then resized to $672 \times 384$; this augmentation expands the dataset from 1,245 to 3,735 samples. Each sample contains 64 consecutive frames at 30 fps, and we split the data into training and validation sets with a 9:1 ratio. Every sample provides three synchronized streams: (i) real robot video $V_{gt}$ (used only as the target path during training for noise/denoise supervision), (ii) simulated foreground $V_{sim}$ rendered in RobotWin by replaying the joint trajectory $q$ with a URDF and camera intrinsics/extrinsics aligned to the real setup, and (iii) background $V_{scene}$ obtained by projecting the simulated silhouette onto $V_{gt}$ and removing the foreground after dilation (kernel size 5, 3 iterations). Instruction text is encoded online by the T5 encoder bundled with CogVideoX-5b-I2V (max length 226, `clean_prompt=True`, `with_attention_mask=True`, `with_cache=True`). The model is built upon `THUDM/CogVideoX-5b-I2V`: the video VAE (AutoencoderKLCogVideoX) is frozen and only encodes videos to the latent space, while the 3D DiT (CogVideoXTransformer3DModel) is the trainable backbone. During training, the target latent is noised at a random timestep, whereas $z_{scene}$ and $z_{sim}$ remain clean as conditions; the three are concatenated along channels in the fixed order $[\tilde{z}_{tar,t}, z_{scene}, z_{sim}]$ and fed to the DiT (input channels = 48), together with the instruction embedding and 3D rotary positional embeddings. The loss is the latent-space diffusion objective implemented by `CogVideoXLoss` (noise/residual prediction). We optimize with AdamW (learning rate $2 \times 10^{-5}$, weight decay $1 \times 10^{-4}$) under a constant schedule, using `bf16` precision and DeepSpeed ZeRO-2 on 4 GPUs (batch size per GPU = 2, gradient accumulation = 8). Training runs up to 100 epochs with EMA and activation checkpointing on `CogVideoXBlock`; checkpoints are saved every 10 epochs with a maximum of 10 kept, and logging uses TensorBoard. Human videos or Grounded-SAM2 segmentation are not used during training; at inference, a human-background video and the IK-replayed simulation serve as conditions to synthesize pseudo-robot videos for downstream VLA training.

**VLA Training**    We train the VLA policy by *mixing* pseudo-robot data from H2R ALIGNER with real demonstrations in a single dataloader. Each sample is a 64-frame window at 30 fps and $672 \times 384$ resolution, paired with the instruction text and time-aligned 14-DoF actions. The model is initialized from **pi0** pretrained weights via the provided `WeightLoader`; parameters selected by `freeze_filter` are frozen (cast to `bfloat16`), while those matching `trainable_filter` are optimized. Training uses the model's built-in behaviour cloning objective `compute_loss` on `(observation, actions)`, optimized with Optax (created by `create_optimizer`) under the configured learning-rate schedule; gradients are computed only over trainable parameters (via `nnx.DiffState`). We run on a multi-device sharded mesh (FSDP) with batch size divisible by device count, enable mixed precision (`bf16`), and maintain EMA weights when `ema_decay` is set. Checkpoints are saved at the configured `save_interval` (with resume support), and Weights&Biases logs loss, gradient norm, and parameter norm at `log_interval`.

## B.2    VISUAL TRANFERRED RESULTS OF *MimicDreamer*

Figure 6 illustrates visual transfer results of *MimicDreamer*. On the left, we show egocentric human demonstration frames for four representative tasks (`Clean Surface`, `Pick up a Bag`, `Insert Tennis`, `Stack Cups`). On the right, we present the corresponding synthesized robot-domain videos generated by *MimicDreamer*, which preserve the task semantics while replacing human hands with robot arms. Additional examples are provided in the supplementary materials.

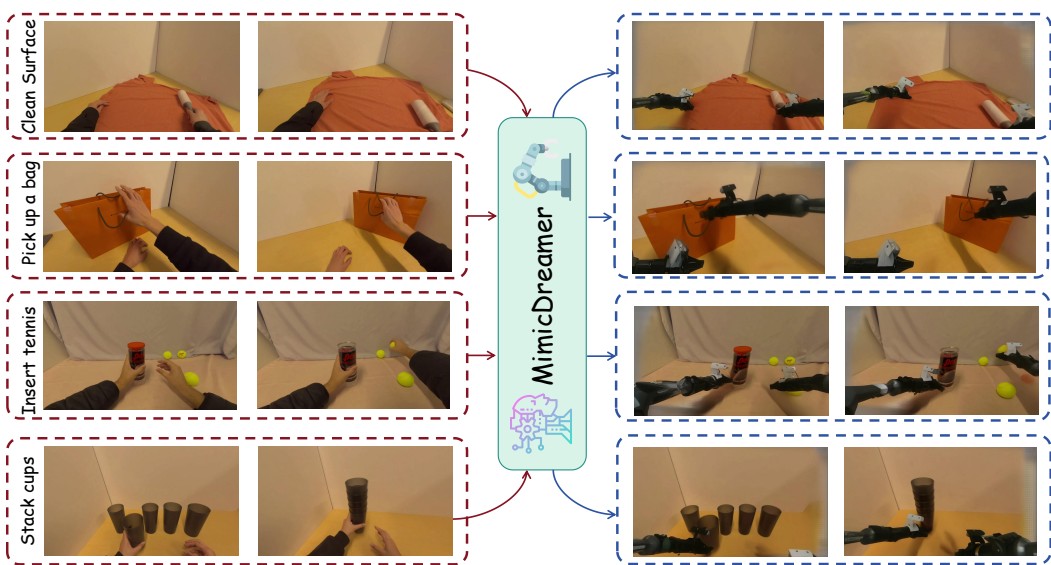

Figure 6: Illustration of videos generated by *MimicDreamer* for human-to-robot transfer, which stabilize egocentric viewpoints and translate human hands into robot manipulators, enabling control of foreground and background appearance while preserving 3D structure and kinematic plausibility.

### B.3 TASK DESCRIPTION

We constructed six scenarios that resemble those in the EgoDex dataset. As shown in Figure 7, we provide the initial state and steps of six tasks. These scenarios are designed to assess a variety of manipulation skills that robots must perform. The details of tasks and corresponding sub-tasks are as follows:

**Pick Bag** Under a neutral background, a robot manipulator interacts with an orange shopping bag on a tabletop. The task is divided into three sub-tasks: **Step 1** Grasp the handle: the end-effector closes to securely hold the bag. **Step 2** Lift and place: the bag is lifted in a stable manner.

**Clean Surface** The manipulator uses a lint roller to clean a blue T-shirt placed on the table. The task contains two sub-tasks: **Step 1** Grasp the roller: the end-effector securely holds the lint roller. **Step 2** Coverage rolling: perform back-and-forth rolling to clean the garment.

**Stack Bowls** Three bowls are arranged on the table with space reserved for stacking. The task contains two sub-tasks: **Step 1** Place the left bowl on top of the middle bowl. **Step 2** Place the right bowl on top of the middle bowl to complete a three-bowl stack.

**Dry Hands** The robot uses its right arm to grasp a towel and wipe its left arm. The task contains two sub-tasks: **Step 1** Grasp the towel: the end-effector securely pinches and holds the towel. **Step 2** Coverage wiping: the towel contacts the left arm, and a wiping motion is executed.

**Insert Tennis** Pick up a tennis ball from the table and place it into the bottle opening. The task contains two sub-tasks: **Step 1** Pick up the ball: the end-effector securely grasps and lifts the ball. **Step 2** Insert into the bottle: move to the bottle mouth and release the ball.

**Stack Cups** Stack two or three cups on a flat surface. The task contains two sub-tasks: **Step 1** Grasp the cup: the target cup for stacking is securely grasped. **Step 2** Stack the cups: all cups are successfully stacked together.

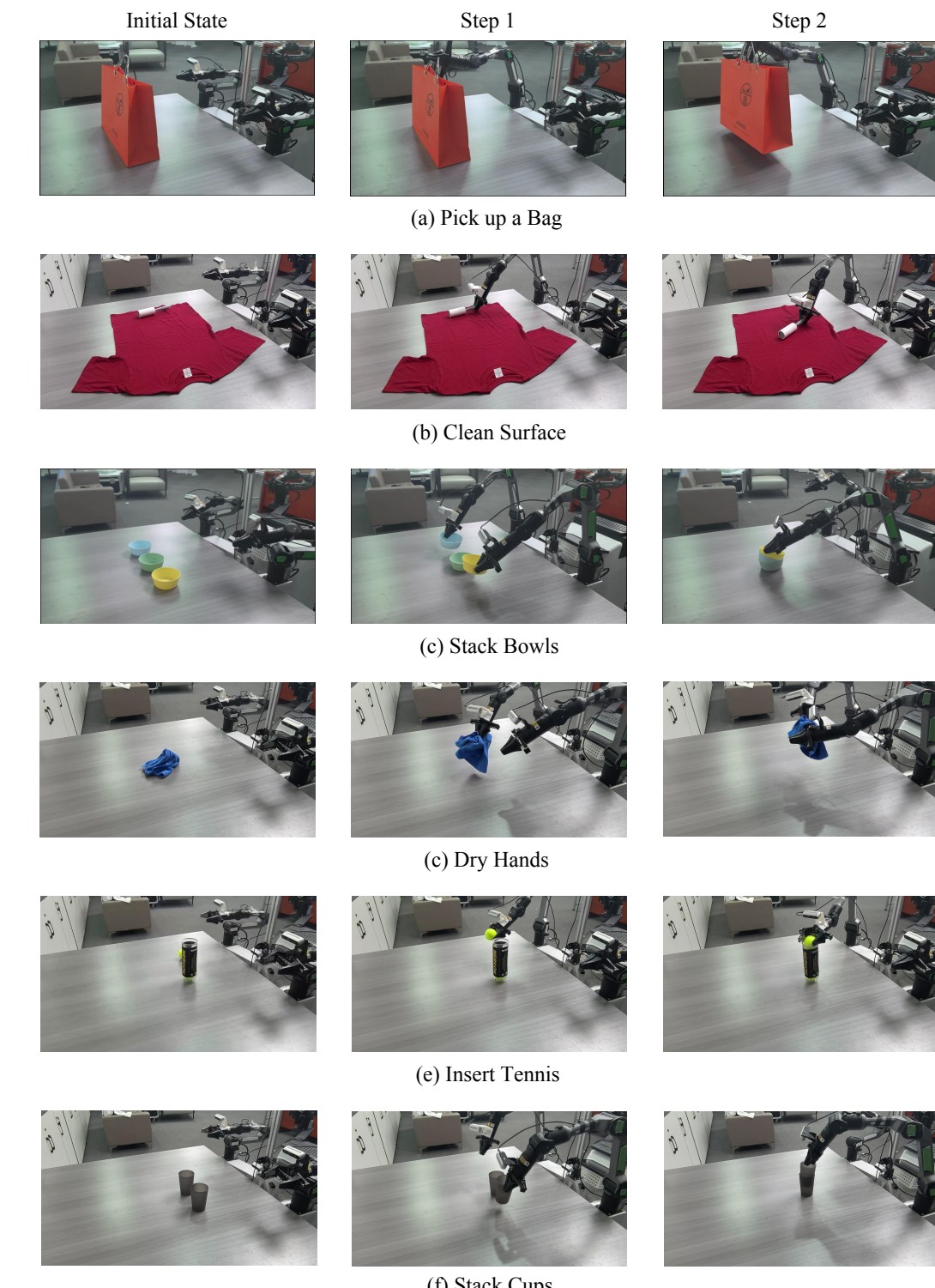

Figure 7: Initial State and Steps of Tasks

## B.4 EVALUATION METRIC FORMULAS

**Notation.** Let $I_t \in \mathbb{R}^{H \times W}$ denote the $t$-th frame (grayscale or luma). Feature correspondences (RANSAC inliers) between two frames are $\{(\mathbf{x}_i, \mathbf{x}_i')\}_{i=1}^N$ with homogeneous coordinates

$\tilde{\mathbf{x}} = [x, y, 1]^\top$. A homography $H_t$ or an affine transform $A_t = \begin{bmatrix} a & b & t_x \\ c & d & t_y \end{bmatrix}$ aligns a source frame to $I_t$. $\Omega_t$ is the valid (inlier/visible) pixel set, with cardinality $|\Omega_t|$. All angles are in degrees ($^\circ$).

**1. View Consistency.** It measures the viewing Angle change (in degrees) of adjacent frames and directly reflects the jitter size The per-step viewpoint change is approximated by the rotation part of $A_t$:

$$\phi_t = \text{atan2}(b, a) \ [^\circ]. \tag{19}$$

Aggregate statistics:

$$\mu_{\text{VC}} = \frac{1}{T-1} \sum_{t=2}^{T} \phi_t, \tag{20}$$

$$\text{P95}_{\text{VC}} = \text{percentile}\big(\{\phi_t\}_{t=2}^{T}, \ 95\%\big), \tag{21}$$

$$\sigma_{\text{VC}} = \sqrt{\frac{1}{T-2} \sum_{t=2}^{T} \big(\phi_t - \mu_{\text{VC}}\big)^2}. \tag{22}$$

**2. Viewpoint Jitter RMS.** It calculates the high-frequency residual energy after the low-pass path, only characterizing "fast jitter". Let $\tilde{\phi}_t = \mathcal{S}(\phi_t)$ be a low-pass filtered version. The jitter energy is

$$\text{JitterRMS} = \sqrt{\frac{1}{T-1} \sum_{t=2}^{T} \big(\phi_t - \tilde{\phi}_t\big)^2}. \tag{23}$$

**3. Homography RMSE (H-RMSE).** We compute the homography reprojection error as a measure of global geometric consistency. With $H_t$ aligning a reference frame (e.g., 1 or $t-1$) to $I_t$, and RANSAC inlier correspondences $\{(\mathbf{x}_i, \mathbf{x}'_i)\}_{i=1}^{N}$ (where $\tilde{\mathbf{x}} = [\mathbf{x}^\top, 1]^\top$ and $\pi([u, v, w]^\top) = [u/w, \ v/w]^\top$), the per-inlier error is

$$e_i = \big\| \pi\big(H_t \, \tilde{\mathbf{x}}_i\big) - \mathbf{x}'_i \big\|_2. \tag{24}$$

The per-frame RMSE is

$$\text{H-RMSE}_t = \sqrt{\frac{1}{N} \sum_{i=1}^{N} e_i^2}. \tag{25}$$

To remove resolution dependence, normalize by the image diagonal $D = \sqrt{W^2 + H^2}$:

$$\text{H-RMSE}_t^{\text{norm}} = \text{H-RMSE}_t / D \ \ \text{(unitless)}. \tag{26}$$

**4. Occlusion-aware MSE.** Warp a reference frame to $I_t$, yielding $\hat{I}_{0 \to t}$. Evaluate only on $\Omega_t$:

$$\text{OccMSE}_t = \frac{1}{|\Omega_t|} \sum_{\mathbf{x} \in \Omega_t} \big(I_t(\mathbf{x}) - \hat{I}_{0 \to t}(\mathbf{x})\big)^2. \tag{27}$$

**5. Dataset-level Aggregation.** For video $v$ with $T_v$ frames and per-frame metric $m_{v,t}$: *frame-weighted* average (recommended for frame-defined metrics):

$$\overline{m} = \frac{\sum_v \sum_t m_{v,t}}{\sum_v T_v}. \tag{28}$$

Alternatively, *per-video equal weight*: compute $\bar{m}_v = \frac{1}{T_v} \sum_t m_{v,t}$, then $\overline{m} = \frac{1}{V} \sum_v \bar{m}_v$.

### B.5 ADDITIONAL QUALITATIVE RESULTS OF EXPERIMENT OF H2R ALIGNER

Figure 8 presents additional qualitative examples of H2R-ALIGNER. Similar to the main paper, each triplet shows the original human demonstration (top), the simulated replay (middle), and the synthesized robot-domain video (bottom). These results further confirm that H2R-ALIGNER consistently produces realistic robot sequences aligned with task semantics and background context.

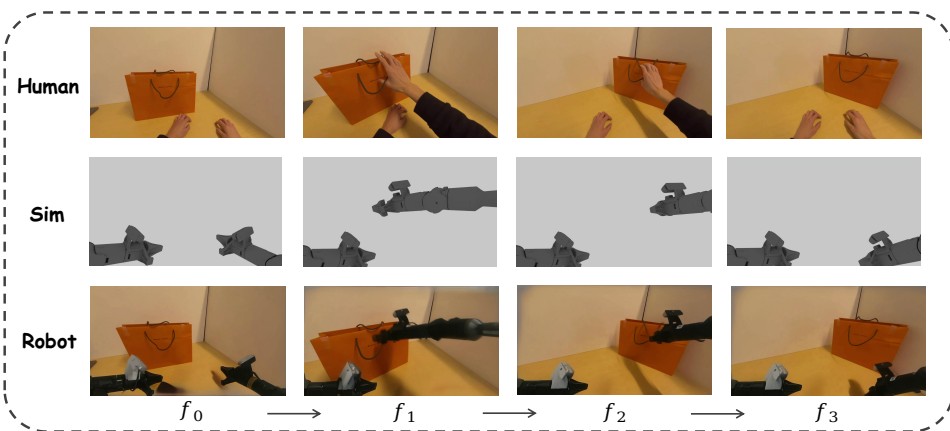

Figure 8: Additional qualitative examples of H2R-ALIGNER. Each triplet shows human demonstration (top), simulated replay (middle), and synthesized robot-domain video (bottom).

Table 3: SR and PSR across tasks as human-to-robot data scale, with 20 robot data trials fixed.

| Setting | Pick Bag | | Clean Surface | | Stack Bowls | | Dry Hands | | Insert Tennis | | Stack Cups | |
|---|---|---|---|---|---|---|---|---|---|---|---|---|
| | SR↑ | PSR↑ | SR↑ | PSR↑ | SR↑ | PSR↑ | SR↑ | PSR↑ | SR↑ | PSR↑ | SR↑ | PSR↑ |
| 20 Robot | 70% | 82% | 90% | 90% | 65% | 80% | 80% | 88% | 25% | 38% | 65% | 80% |
| + 5 Human | 75% | 85% | 95% | 95% | 70% | 85% | 85% | 93% | 25% | 43% | 65% | 85% |
| + 10 Human | 80% | 88% | 97% | 97% | 77% | 88% | 90% | 96% | 30% | 52% | 73% | 87% |
| + 15 Human | 85% | 91% | 98% | 99% | 83% | 90% | 95% | 98% | 37% | 61% | 82% | 89% |
| + 20 Human | 90% | 93% | 100% | 100% | 90% | 93% | 100% | 100% | 45% | 70% | 90% | 90% |
| + 25 Human | 92% | 94% | 100% | 100% | 93% | 94% | 100% | 100% | 48% | 73% | 93% | 95% |
| + 30 Human | 93% | 95% | 100% | 100% | 95% | 95% | 100% | 100% | 50% | 75% | 95% | 95% |

## B.6 MORE RESULTS FOR SCALING EXPERIMENT RESULTS

To quantitatively assess the scalability of our approach, we conducted an experiment where a baseline VLA policy, trained on a fixed set of 20 robot data, was progressively augmented with human-to-robot data. Table 3 presents the results of this analysis, detailing the Success Rate (SR) and Partial Success Rate (PSR) across six manipulation tasks as the number of added human-to-robot data increases incrementally from 5 to 30. This setup allows for a direct evaluation of how performance scales with the quantity of synthesized data while keeping the robot data constant.

The data reveals a clear and consistent trend: performance monotonically improves across all six tasks with the addition of synthesized human demonstrations. This improvement exhibits a "fast-then-steady" scaling pattern, where the most substantial gains in both SR and PSR are typically observed when adding the first 15 to 20 demonstrations. For instance, simpler tasks such as Clean Surface and Dry Hands rapidly approach 100% success, hitting a ceiling effect. Meanwhile, the most challenging task, Insert Tennis, shows the largest relative SR gain (doubling from 25% to 50%), while tasks like Stack Cups demonstrate a significant narrowing of the gap between partial and full success, indicating that the added data effectively refines complex skills.

In summary, these results provide strong empirical evidence for the scalability and effectiveness of our method. This demonstrates that *MimicDreamer* can effectively leverage human input to augment sparse real data, significantly enhancing final policy performance and offering a practical solution to the data scarcity problem in robot learning.

## C    STATEMENT ON THE USE OF LARGE LANGUAGE MODELS

In the preparation of this manuscript, we utilized the large language model (ChatGPT-5 by OpenAI) as a writing assistance tool. Its use was strictly limited to language polishing, which included improving grammar, spelling, clarity, and sentence structure. The LLM was not used for generating scientific ideas, conducting analysis, or interpreting results. The authors have carefully reviewed and edited all model-generated text and take full responsibility for the final content of this paper.

