# OpenReview forum: "MimicDreamer: Aligning Human and Robot Demonstrations for Scalable VLA Training"
_ICLR.cc/2026/Conference — ICLR 2026 Conference Withdrawn Submission_

### Official Review · Reviewer_mmNg · 2025-10-31

**Soundness:** 2
**Presentation:** 3
**Contribution:** 2
**Rating:** 2
**Confidence:** 3

**Summary:**

This paper addresses the problem of translating human demonstration videos into robot training data. The approach involves (1) using retargeting to map human hand poses to robot joint configurations via inverse kinematics (IK), and (2) training a diffusion transformer to synthesize robot observations by conditioning on real background images after removing the human hand and robot appearance rendered using simulation. The resulting policy is co-trained on a mixed dataset comprising both synthetic and real robot data.

**Strengths:**

The idea of converting human demonstration videos into robot training data is intriguing and has the potential to mitigate the high cost and effort associated with collecting in-context robot data.


The proposed method is also evaluated on real robot hardware, and the reported performance appears strong based on the presented results.

**Weaknesses:**

Given the existence of prior works that have already explored retargeting and inpainting for similar objectives [1, 2], the technical contribution of this paper appears relatively incremental.

While one of the main contributions is claimed to lie in the visual generation component, the experiments provided do not adequately evaluate this aspect.

Moreover, the comparative analysis is limited --- more relevant baselines should be included to assess both the quality of the generated visual content and the resulting downstream policy performance.


[1] Phantom: Training Robots Without Robots Using Only Human Videos, CoRL’25;

[2] Masquerade: Learning from In-the-wild Human Videos using Data-Editing, arXiv’25;

**Questions:**

Have the authors performed quantitative evaluations of image realism, such as computing distances in feature space? Furthermore, is there any observed correlation between the realism of the generated images and the downstream policy performance?

Have the authors evaluated the method’s ability to generalize across domains by performing zero-shot transfer, i.e., training the policy entirely with synthetic data and testing it directly in real-world environments?

---

### Official Review · Reviewer_gZL6 · 2025-11-01

**Soundness:** 2
**Presentation:** 3
**Contribution:** 1
**Rating:** 2
**Confidence:** 5

**Summary:**

The paper proposes MimicDreamer, a pipeline to turn cheap human egocentric demos into robot-usable training signals for VLA models by aligning (i) viewpoint (i.e., the EGOSTABILIZER module), (ii) actions (IK-based method to transform the human wrist poses to robot EE poses), and (iii) vision (H2R-ALIGNER diffusion model guided by sim foreground + real background scene videos). On six manipulation tasks, models trained with synthesized human-to-robot data match or outperform robot-only training..

**Strengths:**

The strengths of this paper is:

- Clear writing & coherent system design. The paper is well-structured with a logical flow from problem setup to method and experiments. Each module’s purpose is motivated and the overall pipeline is easy to follow. Figures meaningfully support the text.

- Real-world experiments are convincing. The evaluation includes on-robot tasks with clear metrics, showing consistent gains over baselines.

**Weaknesses:**

Main concern: limited methodological novelty

While the system is thoughtfully engineered, most contributions appear to be system integration of known components rather than new algorithms. Below I outline where prior art already covers similar ideas and where clarifications/ablations are needed.

- Viewpoint Stabilization

  - The egocentric video stabilization pipeline relies on standard video-processing techniques; the value is primarily engineering and integration.

  - The inpainting step depends on high-quality binary masks. Please discuss scalability across diverse scenes/objects and report failure cases (e.g., thin tools, fast motion) and the annotation cost or automation rate of mask generation.

- Action Alignment

    - Using MANO-driven retargeting for arm/hand control is established; see [1-4]:

    - The paper doesn’t situate its retargeting in this literature; please add a related-work comparison and quantify what is new (e.g., constraints, smoothing, calibration, or robustness claims).

    - Most existing work note residual action gaps after retargeting, which is a reason they introduce robot data for co-training [4], or online/offline RL finetuing[2, 3]. Though this work focuses on VLA fine-tuning, how severe such action gap is?

[1]. Qin, Yuzhe, et al. "Anyteleop: A general vision-based dexterous robot arm-hand teleoperation system." arXiv preprint arXiv:2307.04577 (2023).

[2]. Shaw, Kenneth, Shikhar Bahl, and Deepak Pathak. "Videodex: Learning dexterity from internet videos." Conference on Robot Learning. PMLR, 2023.

[3]. Chen, Zoey Qiuyu, et al. "Dextransfer: Real world multi-fingered dexterous grasping with minimal human demonstrations." arXiv preprint arXiv:2209.14284 (2022).

[4]. Liu, Yangcen, et al. "Immimic: Cross-domain imitation from human videos via mapping and interpolation." arXiv preprint arXiv:2509.10952 (2025).

- Visual Alignment: key questions and missing comparisons:
   - Why a diffusion generator? Prior work shows that rendering robot embodiments via direct MANO→robot pose estimation + image inpainting can suffice for single and bimanual tasks; see [5, 6]. Other simple strategies—e.g., inpainting guidance lines or masking both human/robot hands—also reduce visual mismatch [7, 8]. Please include baselines against these simpler, cheaper methods to justify the added complexity.

   - Why condition on sim renders? If the model already sees the real manipulation video (rich scene cues) and the retargeted robot actions, what additional signal do sim-rendered frames provide—geometry? lighting? pose disambiguation? An ablation with/without sim-conditioning (and with IK-replay only) would clarify its contribution. If the goal is merely to convey the robot’s appearance, then straightforward compositing/inpainting—as in [5,6]—should suffice.

[5]. Lepert, Marion, Jiaying Fang, and Jeannette Bohg. "Phantom: Training robots without robots using only human videos." arXiv preprint arXiv:2503.00779 (2025).

[6]. Lepert, Marion, Jiaying Fang, and Jeannette Bohg. "Masquerade: Learning from in-the-wild human videos using data-editing." arXiv preprint arXiv:2508.09976 (2025).

[7]. Kareer, Simar, et al. "Egomimic: Scaling imitation learning via egocentric video." 2025 IEEE International Conference on Robotics and Automation (ICRA). IEEE, 2025.

[8]. Bahl, Shikhar, Abhinav Gupta, and Deepak Pathak. "Human-to-robot imitation in the wild." arXiv preprint arXiv:2207.09450 (2022).

Taking these points together, the work seems better suited to a robotics venue than a learning-focused conference like ICLR.

**Questions:**

- The main question is about the framework novelty instead of integration, see the Weaknesses,

- The minor question: Could lightweight test-time prompting or adapters on off-the-shelf VLA models yield similar gains (e.g., style/camera prompts, action-space remapping), as in [1]?

[1]. Zheng, Ruijie, et al. "Tracevla: Visual trace prompting enhances spatial-temporal awareness for generalist robotic policies." arXiv preprint arXiv:2412.10345 (2024).

---

### Official Review · Reviewer_PTdV · 2025-11-01

**Soundness:** 2
**Presentation:** 3
**Contribution:** 3
**Rating:** 6
**Confidence:** 4

**Summary:**

This paper proposes a framework to directly convert human/egocentric manipulation videos to robot demonstration data. The task is disentangled into two problems. First, for action alignment, the hand trajectory is converted to robot actions with IK solver. Then, this action is utilized to simulate embodiment-only sequences and is sent to a video diffusion model, as well as ego videos processed by stabilization, to produce robot manipulation videos aligning with the egocentric video's environment. The generated videos are used to perform VLA training.

**Strengths:**

1. The paper is well-organized and presented.
2. The problem that the paper is trying to solve is meaningful for scaling up robot data.
3. Distengling action and visual alignment makes sense.

**Weaknesses:**

1. The major concern lies in the robustness/accuracy of the proposed method, and many experiments are missing regarding this issue:
(1) First, the sim robot video is the main input condition of the H2R aligner, so how does the quality of the sim robot video impact the final generation result? What if it is not well-aligned with stable ego videos?
(2) Furthermore, there are many conditions to decide the quality of the sim robot videos, including camera in/ex-trinsics, 3D hand trajectory, and accuracy of IK solver. So what is the actual performance of the hand trajectory extraction, as well as the accuracy of the IK?  More importantly, how does the performance of this impact the accuracy/quality of the derived robot action and its corresponding sim-rendered robot-only videos?  What is the upper bound, for example, to what extent of this factor is wrongly provided, and the system will fail?
(3) What if the real-world egocentric videos have a large gap (camera view, manipulation/grasping points) with the training data?
To summarize, although the reviewer understands that the derived action does not need to be perfect/precise, it is still unknown about the robustness of the system, which is very important for future work.
2. Some related works [1, 2, 3] should be mentioned.

*Minor*:
In the proposed pipeline, the action alignment and ego stabilization are performed first, then they are sent to the H2R aligner. However, the abstract does not clearly show this sequence, which is recommended to be revised.

**Refs:**
[1] Gen2Act: Human Video Generation in Novel Scenarios enables Generalizable Robot Manipulation. CORL 2025.
[2] TASTE-Rob: Advancing Video Generation of Task-Oriented Hand-Object Interaction for Generalizable Robotic Manipulation. CVPR 2025.
[3] Towards Generalizable Zero-Shot Manipulation via Translating Human Interaction Plans. ICRA 2024.

**Questions:**

If the first question of the weaknesses is well-discussed, the reviewer may consider raising the score; otherwise, it will be decreased.

---

### Note · Authors · 2025-11-12

I have read and agree with the venue's withdrawal policy on behalf of myself and my co-authors.